# Validation and psychometric properties of the Brazilian-Portuguese dispositional flow scale 2 (DFS-BR)

Ig Ibert Bittencourt[1]*, Leogildo Freires[2], Yu Lu[3]*, Geiser Chalco Challco[1], Sheyla Fernandes[2], Jorge Coelho[4], Júlio Costa[2], Yang Pian[3], Alexandre Marinho[1], Seiji Isotani[5]

**1** Center of Excellence for Social Technologies/Computing Institute, Federal University of Alagoas, Maceió, Alagoas, Brazil, **2** Institute of Psychology, Federal University of Alagoas, Maceió, Alagoas, Brazil, **3** Advanced Innovation Center for Future Education, Faculty of Education, Beijing Normal University, Beijing, China, **4** Faculty of Medicine, Federal University of Alagoas, Maceió, Alagoas, Brazil, **5** Institute of Mathematics and Computer Science, University of São Paulo, São Carlos, Brazil

* ig.ibert@ic.ufal.br (IIB); luyu@bnu.edu.cn (YL)

## Abstract

Introduction: Flow state is a psychological concept used to describe the optimal engagement in different activities. Therefore, the DFS-2 has been developed as an instrument to measure an individual's dispositional tendency to flow state as a personality trait. Objective: Aiming to obtain an adapted version of the DFS-2 for the Brazilian-Portuguese language (DFS-BR) and for general activities, we performed its forward- and backward-translation, and we validated it. Methods: After gathering answers from 681 Brazilian participants, we performed: (1) the construct validity of the DFS-BR; and (2) the psychometric item quality analysis. Results: the Confirmatory Factorial Analysis (CFA) indicates the best fit for the gathered data is a nine multi-correlated factorial model ($\chi^2/df = 4.23$, $CFI = 0.94$, $TLI = 0.93$ and $RMSEA = 0.069$). Reliability tests performed in this structure indicates excellent internal consistency for the DFS-BR. The item quality analysis indicates that its difficulty and discriminating parameters have a good endorsement to estimate the dispositional flow state. Additionally, we proposed and validated a short version of the DFS-BR (composed of only nine items). The validation results indicates good fit ($\chi^2/df = 2.94$, $CFI = 0.98$, $TLI = 0.97$ and $RMSEA = 0.053$) and good internal consistency. The Test Information Curve of the short version indicates that it is very informative in the estimation of individual dispositional flow state. Discussion and Conclusions: In view of these results, we conclude that the DFS-BR showed good evidence of its validity to be used with Brazilian people. We also suggest the use of a short version when we need only measure the person's flow state based on the principle of Occam's razor. This principle is supported by the analysis presented in this article.

**Data Availability Statement:** All relevant data are within the paper and its Supporting information files.

**Funding:** This work has been supported by the following institutions: Conselho Nacional de Desenvolvimento Científico e Tecnológico (CNPq), Coordenação de Aperfeiçoamento de Pessoal de Nível Superior (CAPES), Fundação de Amparo à Pesquisa do Estado de Alagoas (FAPEAL), and Beijing Normal University (BNU).

**Competing interests:** No authors have competing interests.

## Introduction

The flow experience is the ideal psychological concept associated with high levels of performance [1, 2]. This concept references the entire state of concentration that the people experience when performing activities with a considerable consistency of positive responses [3]. Csikszentmihalyi states that the flow experience is a state in which people have an enjoyable experience, and in which nothing more matters than the activity being doing [4].

A set of internal and external conditions should happen to achieve the flow state [1, 4]. The internal conditions consist of the individual's involvement with the task, a feeling of well-being caused by the task's performance, and a tendency to abstract from everyday life while performing the activity. External conditions include the presence of clear objectives and feedback and the balanced relationship between challenge and skill during the performing of activities. Based on these internal and external conditions, a model composed of nine components was proposed to define the flow experiences.

Employing the model of nine components, in 1996 [5], Jackson and Marsh developed the first version of a structured questionnaire as a self-reported psychometric instrument to measure the flow in sports activities. This instrument is known as the Flow State Scale (FSS), and it comprises the following nine factors. Since the first publication of the FSS, it was continuously adapted and improved by Jackson and Eklund [6], and, nowadays, there are two versions of this psychometric instrument. Both, known as FSS-2 and DFS-2, comprise 36 items with four items for each factor. The instrument DFS-2 is an adapted version of FSS-2, and it was implemented under the assumption that the flow state can be measure as as a *dispositional flow state*, as an individual personality trait that is stable over time, and that defines the dispositional tendency to achieve the flow state [7]. Additionally to the long versions (with 36 items) of the DFS-2 and FSS-2, the authors also defined short versions of both instruments in which one item is defined for each one of nine components in the flow scale. These short versions have been proposed to be used in situations where the use of long instruments is not adequate (e.g., a time limit to answer a questionnaire and difficulty gathering responses).

In 2008, Jackson, Martin, and Eklund provided evidence that corroborated the multidimensional model of the DFS-2 and FSS-2 [8]. Since then, these instruments in both versions, long and short, have been validated for different languages and different physical activities, presenting themselves as a good consistent cross-cultural psychometric instruments [9–11]. The DFS-2 and FSS-2 have been also validated for other diverse activities, such as learning [7, 12], leisure [13], musical performance [14], and online games [15].

Despite the possibility to buy and obtain a translated version of the DFS-2 for the Portuguese-Brazil language (from the website mindgarden.com). Few published studies have been conducted in Brazil to demonstrate its validity. When a psychometric instrument is translated to a new language and introduced in a new cultural context, for its validation, ensuring its conceptual and idiomatic equivalence is necessary [16, 17]. Translation of items needs to maintain the same equivalence in what are measured and operationalized in its original language, and the adaptation for a new culture should consider the relevance of concepts and the domains in which the instrument is applied [18]. Thus, it is necessary to evaluate the appropriateness of each item in terms of the ability of respondents to represent the concepts measured by the instruments in different domains. In this sense, various validation studies should be conducted in the new culture, in different domains (including, general activities), with different populations, and using different versions of the same instruments (this last for cross-cultural validation).

In Brazil, studies to validate the Portuguese-Brazilian version of the DFS-2 were conducted only in few specific domains, such as the work of Freitas et al. (2019) [19], and the sport and

physical activities [20–25]. Only, a recent study conducted by Correia et al. (2020) [26] performed the validation of the DFS-2 in the Brazilian-portuguese language for general activities. However, this study did not perform validation of the short version of DFS-2, and only classical validation methods (construct validity and internal consistency) were carried out. For these reasons, and to complement results from previous validations, we conducted a study (reported in this paper) in which, in addition to the classical validation methods, we performed a psychometric item quality analysis of the instrument using Item-Response Theory (IRT). We also proposed and conducted a validation of the short-version of the DFS-2 for the Portuguese-Brazil language.

## Methods

### Participants

We employed a non-probabilistic sample (by voluntary sampling) of $n$ = 681 Brazilians as participants in this study. All of them, from 18 to 74 years old, with an average age of $M$ = 27.17 years old, and $SD$ = 12.29.

A self-reported socio-demographic questionnaire was answered by the participants to gather information of their gender, civil status, ethnicity, socio-economic status, and sexual orientation. The majority of participants were men 59.03%, 40.09% were female, and 0.88% did not declare their gender. The civil status of 70.63% participants was single, 20.12% were married, 6.75% were cohabiting with a partner, 2.06% were divorced, and 0.44% were widow. Regarding their ethnicity, 61.81% declared to be white, 23.20% declared to be pardo (mixed-race), 7.64% declared to be black, 2.06% declared to be yellow (mongoloid asian), 0.59% declared to be part of indigenous tribal population, and 4.70% preferred not to declare their ethnicity. Social economic status of participants were: 46.70% of the middle class, 32.89% of middle lower class, 9.69% of lower class, 8.81% of middle upper class, and 1.91% of higher economic class. Sexual orientation of participants: 83.26% declared themselves as heterosexual/ straight, 5.29% declared to be bi-sexual, 4.11% declared to be homosexual, 0.15% declared to be transsexual, and 7.20% did not wish to declare their sexual orientation.

In order to be participants of this study, respondents needed to declare themselves as Brazilian citizens, residing in Brazil, and having fluency in Portuguese language. Inconsistent answers were removed from the data, and the respondents of these answers were not considered part of participants.

In the questionnaire used to gather the data, respondents were also indicated the activity for which the dispositional flow state was measured. These activities were: learning about physics (33.14%), security information (24.93%), and other 192 activities (41.93%, with less than 3% per activity).

### Recruitment

We performed the data collection entirely through the Internet, using volunteer participation, in which the involved researchers sent recruitment messages by their own social media networks (e.g., Facebook, Instagram, Whatsapp) and email to obtain responses to the questionnaires used in this study. No one specific groups were targeted in the social media networks. After reading the instructions, the participants answered the questionnaire available in electronic form on the website: http://flow.lyralemos.com.br/. When the filling of the questionnaire was interrupted, the respondents were informed, and they could resume the questionnaire later as long as they did not delete their browsing history. The form was available for 90 days, the needed time to achieve the sample employed in this study. During this time, the involved

researchers also made themselves available to clarify any doubts to the participants regarding the study.

## Ethics approval

We strictly followed all the prerogatives in the resolutions 466/12 and 510/16 of the Brazilian National Health Council (CNS). Therefore, the Human Research Ethics Committee of the Federal University of Alagoas (with Protocol n. 35701820.3.0000.5013) approved this research study. We informed the participants that they were not obligated to collaborate with the research, and that they could, at any time, decline their participation whether they felt any discomfort for any reason. Before answering the questionnaire, the participants agreed with a Free Prior and Informed Consent (FPIC) in which we indicated to the participants that their provided information would be confidential, without any possible individual identification, and that their responses being only analyzed as a whole and not individually. On average, reading the FPIC and answering the questionnaire took twenty (20) minutes for each participant.

## Procedure

The translation and adaptation of the DFS-2 for the Brazilian context were carried out according to the procedures of International Test Commission [27]. Two independent interpreters performed the forward translation of items from English to Brazilian-Portuguese. Subsequently, a backward translation was carried out by a third independent translator from Portuguese to English. Then, comparing the original and back-translated versions of the instrument, it was concluded that they were semantically equivalent. Finally, we performed a semantic validation of DFS, we checked the adequacy of the level of understanding and understanding of the items with the target population. Following this procedure, we reached the initial version of the DFS-BR instrument (see S1 Appendix).

## Instruments

The DFS-BR comprises 36 items (see S1 Appendix) based on the original DFS developed by Jackson and Eklund (in 2002) [6], revised by Jackson, Martin, and Eklund (in 2008) [8], and detailed in the manual published in 2010 [2]. In addition to the 36 items, we gathered demographic information of respondents such as age, gender, sexual orientation, marital status, and social-economic status. The DFS-BR employed a rating scale measure of the 5-point Likert scale, ranging from 1 (never) to 5 (always), with four items for each one of the following nine components (also known as factors and dimensions): Challenge-Skill Balance (CSB), Merging of Action-Awareness (MAA), Clear Goals (CG), Unambiguous Feedback (UF), Concentration on Task at Hand (CTH), Sense of Control (SC), Loss of Self-Consciousness (LSC), Transformation of Time (TT), and Autotelic Experience (AE).

**Data analysis procedure.** Firstly, we conducted the construct validity of both versions of the DFS-BR, in their structure, internal consistency, convergence, and discrimination. This validation assesses whether the instruments really measure the latent psychological concepts intended to be measured. In our case, this concept is the dispositional flow state, and its nine components. After this validation, we performed the psychometric item quality analysis of both versions using IRT. We used the R program [28] to conduct all the validation and analysis.

Incomplete responses were automatically removed from the dataset by the web application in which the respondents answered the items of the DFS-BR. The Dataset employed to perform the analyses is available in S1 Dataset.

**Table 1. Descriptive statistics, normality test, and MSA of the dataset.**

|  | M | SD | Mdn | Skew | Kurtosis | Statistic | p.val | MSAi |
|---|---|---|---|---|---|---|---|---|
| i1 | 3.737 | 0.967 | 4 | -0.635 | 0.252 | 0.8714 | <0.001 | 0.970 |
| i2 | 3.562 | 0.904 | 4 | -0.508 | 0.448 | 0.8707 | <0.001 | 0.961 |
| i3 | 3.106 | 0.978 | 3 | -0.165 | -0.190 | 0.9017 | <0.001 | 0.959 |
| i4 | 3.637 | 0.931 | 4 | -0.621 | 0.321 | 0.8711 | <0.001 | 0.959 |
| i5 | 3.470 | 0.931 | 4 | -0.403 | 0.013 | 0.8872 | <0.001 | 0.958 |
| i6 | 3.079 | 1.083 | 3 | -0.053 | -0.663 | 0.9159 | <0.001 | 0.965 |
| i7 | 3.523 | 0.898 | 4 | -0.549 | 0.426 | 0.8701 | <0.001 | 0.955 |
| i8 | 3.003 | 1.183 | 3 | 0.063 | -0.822 | 0.9156 | <0.001 | 0.894 |
| i9 | 3.555 | 1.012 | 4 | -0.422 | -0.143 | 0.8914 | <0.001 | 0.827 |
| i10 | 3.752 | 0.987 | 4 | -0.651 | 0.110 | 0.8723 | <0.001 | 0.953 |
| i11 | 3.485 | 0.936 | 4 | -0.648 | 0.448 | 0.8674 | <0.001 | 0.962 |
| i12 | 2.883 | 0.934 | 3 | -0.058 | -0.359 | 0.8994 | <0.001 | 0.892 |
| i13 | 2.925 | 0.951 | 3 | -0.025 | -0.320 | 0.9030 | <0.001 | 0.884 |
| i14 | 3.941 | 0.931 | 4 | -0.833 | 0.617 | 0.8450 | <0.001 | 0.934 |
| i15 | 3.537 | 0.894 | 4 | -0.415 | 0.081 | 0.8798 | <0.001 | 0.950 |
| i16 | 3.339 | 0.993 | 3 | -0.339 | -0.198 | 0.8994 | <0.001 | 0.931 |
| i17 | 3.380 | 0.901 | 3 | -0.242 | -0.052 | 0.8888 | <0.001 | 0.963 |
| i18 | 2.786 | 1.160 | 3 | 0.270 | -0.690 | 0.9102 | <0.001 | 0.908 |
| i19 | 3.501 | 1.035 | 3 | -0.356 | -0.206 | 0.8926 | <0.001 | 0.935 |
| i20 | 3.718 | 0.945 | 4 | -0.550 | 0.217 | 0.8732 | <0.001 | 0.945 |
| i21 | 3.167 | 0.880 | 3 | -0.331 | 0.189 | 0.8772 | <0.001 | 0.947 |
| i22 | 2.814 | 0.948 | 3 | 0.099 | -0.236 | 0.9011 | <0.001 | 0.872 |
| i23 | 3.539 | 0.870 | 4 | -0.594 | 0.459 | 0.8620 | <0.001 | 0.967 |
| i24 | 3.627 | 0.954 | 4 | -0.436 | -0.083 | 0.8859 | <0.001 | 0.935 |
| i25 | 3.699 | 0.869 | 4 | -0.696 | 0.618 | 0.8522 | <0.001 | 0.960 |
| i26 | 3.567 | 0.921 | 4 | -0.461 | 0.109 | 0.8815 | <0.001 | 0.962 |
| i27 | 3.514 | 0.914 | 4 | -0.393 | 0.064 | 0.8841 | <0.001 | 0.977 |
| i28 | 3.072 | 1.204 | 3 | -0.032 | -0.904 | 0.9154 | <0.001 | 0.895 |
| i29 | 3.123 | 1.116 | 3 | -0.079 | -0.594 | 0.9139 | <0.001 | 0.908 |
| i30 | 3.921 | 0.970 | 4 | -0.749 | 0.252 | 0.8525 | <0.001 | 0.955 |
| i31 | 3.567 | 0.923 | 4 | -0.409 | 0.132 | 0.8817 | <0.001 | 0.938 |
| i32 | 3.664 | 0.941 | 4 | -0.567 | 0.210 | 0.8761 | <0.001 | 0.940 |
| i33 | 3.746 | 0.869 | 4 | -0.683 | 0.802 | 0.8527 | <0.001 | 0.967 |
| i34 | 3.229 | 1.226 | 3 | -0.160 | -0.899 | 0.9098 | <0.001 | 0.876 |
| i35 | 3.627 | 1.065 | 4 | -0.552 | -0.151 | 0.8846 | <0.001 | 0.788 |
| i36 | 3.862 | 0.971 | 4 | -0.887 | 0.766 | 0.8463 | <0.001 | 0.951 |

Before performing the construct validity, we also examined the suitability of the dataset to conduct factor analysis using Bartlett's Test of Sphericity and Kaiser-Meyer-Olkin (KMO). The results of these tests indicated that the data is adequate for factor analysis with $\chi^2 =$ 13610.18, $df = 630$ and $p < 0.01$, and with an overall Measure of Sampling Adequacy (MSA) of 0.94 (marvelous). Table 1 details the values regarding the descriptive statistics (M: mean, SD: standard deviation, and Mdn: median), normality tests (Skewness, Kurtosis, Statistic and p-value of Shapiro-Wilk test), and the MSA for each one of items (MSAi) used in this study.

Lavaan package [29] was used to conduct a Confirmatory Factor Analysis (CFA), and to determine the structure and item loadings in both versions of the DFS-BR. Due to the ordinal

nature of the data, we used Weighted Least Squares Mean and Variance-Adjusted (WLSMV) estimator with polychoric matrices for performing the CFA. WLSMV estimator does not require the normality distribution in the observed data to estimate the fit indexes [30, 31]. Polychoric correlation reduces the effect of attenuation that occur in the measured latent variable when it used a small number of item levels to estimate this value [32], and it is frequently applied for self-report psychometric instruments, such as personality trait [33].

For the long version of the DFS-BR, we tested two models: (1) the multi-correlated model of nine factors [8]; (2) the second-order model of nine factors grouped by one second-order factor [7]. There was no need to establish fixing parameters, starting values, modifiers, or error values in the models. We only fixed to 1 the interfactor correlations of factors with values greater than 1 because WLSMV yielded to give a moderate overestimation for this parameter [31].

Results from the CFA was analyzed using the following fit indexes and goodness-of-fit criteria: (a) for the chi-square/degrees of freedom ($\chi^2/df$), values less than 2 indicated excellent fit, values between 2 and 3 were a good fit, and values up to 5 were acceptable [34]; (b) for the Comparative Fit Index (CFI) and the Tucker-Lewis index (TLI), values higher or close than 0.90 were considered acceptable; (c) for the Goodness-of-Fit Index (GFI) and the Adjusted Goodness-of-Fit Index (AGFI), that takes into account the degrees of freedom of the model with respect to the number of variables, values close or greater than 0.90 were considered acceptable [34, 35]; (d) for the Standardized Root Mean Square (SRMR) and the Root Mean Square Error of Approximation (RMSEA), as error between the model and the observed data, values until to 0.10 were considered admissible, values lower than 0.08 were acceptable, and values lower than 0.05 indicated a good fit [36, 37].

The internal consistency was evaluated employing the semTool package [38], and in addition to calculate the classical reliability test of Cronbach's alpha ($\alpha$), we also calculated the McDonald's omega ($\omega$) [39] and Composite Reliability (CR) [40]. These latter both reliability measures overcome the underestimation of Cronbach's $\alpha$ caused by the assumption of tau-equivalences. McDonalds' $\omega$ and CR use congeneric models, so the factor loadings may vary in all the items for the factorial structure. For these reliability tests, we defined 0.60 as the cut-off value. Thus, values close or above 0.60 were considered acceptable, values above 0.80 were excellent, and values of 0.70s indicated good reliability.

Regarding the convergent and discriminant validity, we evaluated the values of CR and the Average Variance Extracted (AVE) [41], as well as the heterotrait-monotrait (HTMT) ratio [42]. These indexes were calculated using the semTools package [38], and for the convergent validity, we adopted the following criteria: AVE with values higher than 0.5 indicated good convergence of factors [43]; the convergence was still adequate if AVE is less than 0.5, but the CR of all factors are greater than 0.6 [41]. For the discriminant validity, we examined the AVE values and the correlations of factors. According to the literature [44], there is a strong discriminant of factors whether the square root of the AVEs are greater than the correlation coefficients between the factors. This method usually has been proved with a high false positive rate, indicating discriminant problems under conditions where there are no real issues [45]. Thus, we complemented the evaluation of AVE and the correlations factors with the assessment of HTMT ratio. Based on the criteria HTMT0.85 [46], values greater than 0.85 indicate problems in the discriminant validity. Sometimes, these problems are caused by the presence of multicollinearity problems in the gathered data, so we calculated the Variance Inflation Factor (VIF) of factors to check this assumption.

For the analysis of the psychometric item quality in both versions of the DFS-BR, we employed the IRT analysis. It is a framework to assess psychometric instruments by explaining the dependencies between item responses within a person and between persons [47]. IRT has

been developed to fill gaps of the Classical Test Theory (CTT), such as the impossibility of assessing the individual parameters for each item, and the unrealistic assumption that there is uniformity of confidence intervals for a person's latent construct [48]. In this study, the dispositional flow state is the latent construct measured by the instrument, and the use of IRT leads us to have a more realistic confidence intervals for this measure wherein there are different levels of item parameters used for different levels of dispositional flow state. These parameters are the difficulty and discrimination of items. The item difficulty, based on the item responses within all participants, determines the manner in which the item behaves along the measured scale, so that this parameter in each item level is an estimate of the dispositional flow state level to pass this level. Item discrimination indicates the endorsing degree for a correct item level given a latent constructor level, so this value determines the quality of item to differentiate similar levels of the latent construct being measured by the psychometric instrument.

For the IRT analysis, we employed the Rating Scale Model (RSM) [49] to determine the parameters of difficulty and discrimination in the DFS-BR. We used the Mirt package [50] to calculate these parameters. Based on Baker (2001) [51], the item discrimination power was classified in: very-low (0.01 to 0.34), low (0.35 to 0.64), moderate (0.65 to 1.34), high (1.35 to 1.69), and very high (>1.70) discrimination. To classify the parameter of item difficulty, we employed the classification proposed in [52] in which the item difficulty vary between −3 and + 3. Thus, when the difficulty item is less than −1.28, it is considered very-easy; between −1.28 and −0.53, it is classified as easy; between −0.52 and 0.52, it is considered as moderate; between 0.53 and 1.28, it is considered as difficult; and greater than 1.28, it is considered very-difficult. Besides the estimation of difficulty and discriminant parameters, we evaluated Test Information Curves (TIC) to depict the intervals in which our instrument is more accurate to yield any ability level [53].

## Results

### Factor structure validity

Table 2 summarizes the fit indexes for the two tested models in the CFA. The indexes of $\chi^2/df$ (between 3 and 5), CFI (above 0.90), TLI (above 0.90), GFI (above 0.90), AGFI (above 0.90), and SRMR (between 0.05 and 0.10) indicated that both model are acceptable, and the RMSEA values indicate a good fit (close to 0.05) with the gathered data. To assess whether there are significant differences between both models and the gathered data, we performed an ANOVA test, and the result indicates that there is a significant difference with $\Delta\chi^2(10) = 340.73$, and $p < 0.001$.

Based on the fit indexes and significant differences, the multidimensional correlated model fits better for our empirical gathered data. Fig 1 shows the item loadings for this model in which the nine factors are: Challenge Skill Balance (CSB); Autotelic Experience (AE); Transformation of Time (TT); Lost of Self-Consciousness (LSC); Sense of Control (SC); Concentration on the Task at Hands (CTH); Unambiguous Feedback (UF); Clear Goals (CG); and Merging of Action and Awareness (MAA). All the items with the exception of the item i35 have loadings greater than 0.30 which is considered adequate according the criteria defined by Hair et al. (2013) [54], for sample size of more than $n > 350$. Item i35 is not removed to

**Table 2. Fit indexes of tested models in the validation of DFS-BR.**

|  | $\chi^2$ | df | $\chi^2/df$ | CFI | TLI | GFI | AGFI | SRMR | RMSEA | RMSEA.CI |
|---|---|---|---|---|---|---|---|---|---|---|
| multicorrelated | 2431.181 | 575 | 4.228 | 0.938 | 0.932 | 0.944 | 0.935 | 0.091 | 0.069 | [0.066; 0.072] |
| 2nd-order | 2541.138 | 585 | 4.343 | 0.934 | 0.929 | 0.941 | 0.933 | 0.093 | 0.070 | [0.067; 0.073] |

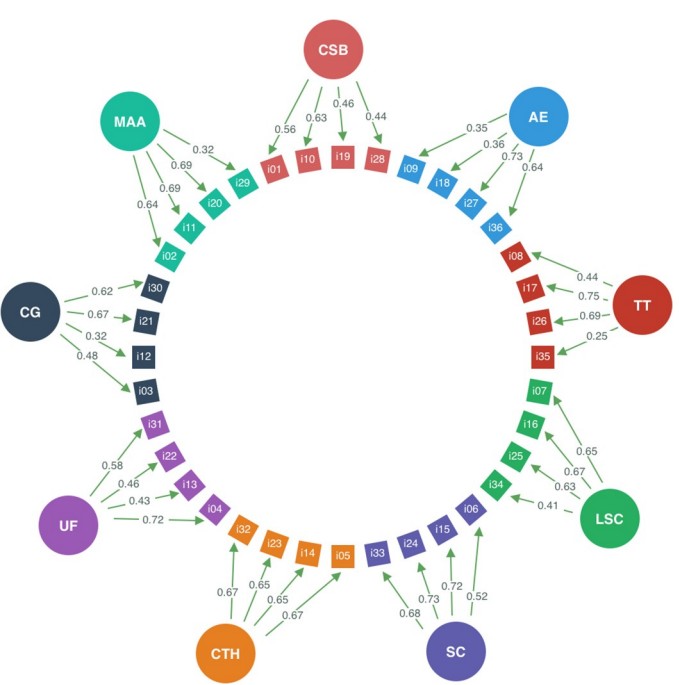

**Fig 1. Item loadings in the nine multi-correlated model of the DFS-BR.**

maintain congruence with the original version of the DFS-2 and the theoretical underlie of the flow theory.

It is important to highlight here that, in the multi-correlate model (Fig 1), at least one item has close or greater values than 0.70 in the item loadings. In the development of the original version of the DFS-2 [2], they were used to define the short version of the DFS-2. In our study, the items with these high values are the item i10 for the CSB. For the MAA, these items are i11 and i20. For the CG, these items are i21 and i30. For the UF, this item is i4. For the CTH, all the four items i5, i14, i23 and i32 are close to 0.70. For the SC, the items with good loadings are i15 and i24. For the LSC, the items with the best loadings are i7 and i16. For the TT, the item with factorial loadings of 0.70s was the i17. For the AE, the item i27 and the item i36 have good loadings.

Table 3 details the matrix of factor covariances found in the nine multi-correlated model. The covariances indicates that there is no one weak linear relationship (<0.50) for the factors,

**Table 3. Covariances in the nine factor model of the DFS-BR.**

|  | **CSB** | **MAA** | **CG** | **UF** | **CTH** | **SC** | **LSC** | **TT** | **AE** |
|---|---|---|---|---|---|---|---|---|---|
| CSB | - |  |  |  |  |  |  |  |  |
| MAA | 1 | - |  |  |  |  |  |  |  |
| CG | 1 | 1 | - |  |  |  |  |  |  |
| UF | 0.846 | 0.896 | 1 | - |  |  |  |  |  |
| CTH | 0.867 | 0.923 | 0.814 | 0.932 | - |  |  |  |  |
| SC | 0.844 | 0.876 | 0.816 | 0.934 | 1 | - |  |  |  |
| LSC | 1 | 0.908 | 0.856 | 0.87 | 1 | 1 | - |  |  |
| TT | 1 | 1 | 0.838 | 0.843 | 1 | 1 | 1 | - |  |
| AE | 1 | 1 | 0.885 | 0.82 | 0.889 | 0.893 | 1 | 1 | - |

**Table 4. Results of the internal reliability tests for the DFS-BR.**

|  | CSB | MAA | CG | UF | CTH | SC | LSC | TT | AE | total |
|---|---|---|---|---|---|---|---|---|---|---|
| $\alpha$ | 0.563 | 0.628 | 0.594 | 0.660 | 0.752 | 0.733 | 0.643 | 0.546 | 0.555 | 0.940 |
| $\omega$ | 0.600 | 0.666 | 0.609 | 0.639 | 0.758 | 0.755 | 0.667 | 0.601 | 0.591 | 0.941 |
| CR | 0.610 | 0.686 | 0.612 | 0.640 | 0.758 | 0.764 | 0.691 | 0.632 | 0.612 | 0.948 |

and that there are strong relationship (>0.70) among all the possible pairs. Observing the covariance table, we can identify that there are many pairs with perfect linear relationships (correlation of value 1) which means that these pairs of factors should be combined in only one factor. However, the underpinning theory (flow theory) states that the dispositional flow state depends on nine components. We can also indicate that these covariances support the validity of the 2nd order model in which all nine factors are aggregated in only one factor. This model also has acceptable fit indexes for the empirical data gathered in this study.

## Internal consistency

Table 4 shows the results of reliability tests performed on the DFS-BR. For the composition of all 36 items, the Cronbach's $\alpha$, McDonalds' $\omega$ and CR values were greater than 0.90 indicating an excellent internal consistency to measure the dispositional flow state. Good reliability of internal consistency (0.70s) was observed for the Concentration on the Task at Hand (CTH) and Sense of Control (SC). According to all the reliability tests, there was an acceptable reliability (0.60s) for the Merging Action-Awareness (MAA), Unambiguous Feedback (UF), and Lost of Self-Consciousness (LSC).

Although the Challenge-Skill Balance (CSB), Clear-Goals (CG), Transformation of Time (TT), and Autotelic Experience (AE) present Cronbach's $\alpha$ that are less than the cutoff value (0.60). Their McDonalds' $\omega$ and CR values indicate an acceptable internal consistency (values greater than 0.60). Based on these results, we can conclude that the DFS-BR presents an acceptable internal consistency for the nine components of the flow theory, and an excellent internal consistency to measure the individuals' dispositional flow state.

## Convergent validity and discriminant validity

Table 5 shows the CR, AVE, VIF, square root of AVEs, factor correlations, and HTMT ratios. As we explained in the data analysis procedure, these values are indexes used to evaluate the

**Table 5. Results of the convergent and discriminant validity of the DFS-BR.**

|  | CR | AVE | VIF | CSB | MAA | CG | UF | CTH | SC | LSC | TT | AE |
|---|---|---|---|---|---|---|---|---|---|---|---|---|
| CSB | 0.610 | 0.274 | 3.092 | **0.524**[*] | 1.196 | 1.059 | 0.874 | 0.923 | 0.916 | 1.090 | 1.256 | 1.301 |
| MAA | 0.686 | 0.343 | 3.122 | 1 | **0.585**[*] | 1.113 | 0.923 | 0.959 | 0.935 | 0.983 | 1.137 | 1.146 |
| CG | 0.612 | 0.292 | 2.504 | 1 | 1 | **0.540**[*] | 1.087 | 0.822 | 0.840 | 0.900 | 0.935 | 0.994 |
| UF | 0.640 | 0.315 | 2.353 | 0.846 | 0.896 | 1 | **0.561**[*] | 0.893 | 0.921 | 0.886 | 0.887 | 0.856 |
| CTH | 0.758 | 0.44 | 3.461 | 0.867 | 0.923 | 0.814 | 0.932 | **0.663**[*] | 1.096 | 1.011 | 1.034 | 0.924 |
| SC | 0.764 | 0.436 | 3.829 | 0.844 | 0.876 | 0.816 | 0.934 | 1 | **0.661**[*] | 1.085 | 1.049 | 0.952 |
| LSC | 0.691 | 0.336 | 3.282 | 1 | 0.908 | 0.856 | 0.870 | 1 | 1 | **0.580**[*] | 1.241 | 1.105 |
| TT | 0.632 | 0.291 | 3.474 | 1 | 1 | 0.838 | 0.843 | 1 | 1 | 1 | **0.539**[*] | 1.348 |
| AE | 0.612 | 0.277 | 2.979 | 1 | 1 | 0.885 | 0.820 | 0.889 | 0.893 | 1 | 1 | **0.527**[*] |

[*]: The bold numbers listed diagonally in the right columns of the VIF values are the square root of AVE values. The correlation among the factors are located in the lower part of the square root of AVE values, and the HTMT ratio of the correlation coefficients are located in the upper part of the square root of AVE values.

convergent and discriminant validity of the DFS-BR. Convergent validity is assessed by the item loadings, as well as the CR and AVE values. All the AVE values are less than 0.50, the cut-off frequently defined to ensure convergence. Four of nine factors (UF, SC, TT, AE) have at least one item with high level of convergence (item loadings greater than 0.70), and the CR values indicate acceptable (0.60s) and good (0.70s) convergence for all the factors. Based on these results, we state that the multi-correlation model has an acceptable convergence validity. This statement is based on the structure equation modelling literature in which it is said that the AVE is a very conservative test [55]. A convergent validity is still acceptable when the AVE values are less than the cutoff of 0.50, but all the CR values are greater than 0.6 [41].

Based on Table 5, we can observe that there are no multicollinearity problems because all the VIF values are lower than 5 (cutoff value). If VIF values are greater than 5, and the correlations are greater than 0.80, it is suggested the combination of the factors [54]. However, this suggestion is subject to the underpinning theory. No one VIF value is greater than 5, so that there is no need to combine any factor in the multi-correlated model based on this criterion.

To assess the discrimination validity of the multi-correlated model based on the Fornell-Larcker criterion [41], and the similarity degree between factors [42], the columns to the right of VIF values are composed by the factor correlations (lower triangular part), their HTMT ratio (upper triangular part), and the square root of AVE values (diagonal bold values). According to the Fornell-Larcker criterion, when the square root of the AVE value of each factor is greater than its correlations, it is an indication that the difference between each measurement factor is better [54], demonstrating a good discriminant validity. Table 5 shows that this criterion has not been satisfied for the multi-correlated model. The upper triangular part of this Table shows the HTMT ratios of correlations. With exception of the pairs CG-CTH and CG-SC, all the rest of correlation pairs have HTMT ratios greater than 0.85 indicating discriminant problems for the six factors of Challenge-Skill Balance (CSB), Merging of Action-Awareness (MAA), Unambiguous Feedback (UF), Loss of Self-Consciousness (LSC), Transformation of Time (TT), and Autotelic Experience (AE).

Although the lack of discriminant validity can be understood as overlapping items in the factors of a psychometric instrument, or that the factors are measuring the same thing, in our case, we consider these high HTMT values as a confirmation that the nine factors can be aggregated in only one psychological concept, which is known as the individual's dispositional tendency towards flow state according to the Csikszentmihalyi flow theory [1].

## Construct validity of the DFS-Short BR

Jackson, Martin and Eklund (2008) [8] also developed and validated an alternative Brazilian-Portuguese short version of the DFS-2. This short version, abbreviated as DFS-Short BR (see S2 Appendix), comprises only nine items from all the 36 items used in the long version of the DFS-BR. Aiming to assess the structure validity of the short version, we conducted the CFA employing two unidimensional models. The former model (unidim1) was constituted by the items indicated in the original version of the DFS-2 [2], and the latter model (unidim2) was defined with items that have highest loadings in the nine multi-correlated model. In this alternative model (unidim2), the item i10 replaces the item i19 of the DFS-Short BR to measure the CSB, the item i20 replaces the item i29 to measure the MAA, the item i21 replaces the item i12 to measure the CG, the item i4 replaces the item i22 to measure the UF, the item i24 replaces the item i6 to measure the SC, and the rests of items (items i32, i7, i17 and i36) have been maintained to measure the CTH, LSC, TT and AE.

Notice that the method uses to define the alternative shorten version of the DFS-BR (with the model *unidim2*) increases the reliability of the instrument, but it may cause deterioration

**Table 6. Fit indexes of tested models in the validation of DFS-BR.**

| | $\chi^2$ | df | $\chi^2/df$ | CFI | TLI | GFI | AGFI | SRMR | RMSEA | RMSEA.CI |
|---|---|---|---|---|---|---|---|---|---|---|
| unidim1 | 210.065 | 27 | 7.780 | 0.876 | 0.835 | 0.957 | 0.928 | 0.099 | 0.100 | [0.088; 0.113] |
| unidim2 | 79.500 | 27 | 2.944 | 0.981 | 0.975 | 0.986 | 0.976 | 0.070 | 0.053 | [0.040; 0.067] |

in the validity of the scale. To avoid a high degradation, improving simultaneously the reliability and the fit indexes, we intended to minimize the number of items that were changed in the model *unidim1*. We only changed the original item in the short version when the difference between the highest loading for each factor and the loading of the original item was greater than 0.20 in the multicorrelated model.

Table 6 summarizes the fit indexes of the CFA performed in the two models used to assess the structure validity of the DFS-Short BR. The results show that the model unidim1 does not have adequate good-fitting values for the indexes of $\chi^2/df$ (above 5), CFI (below 0.90), TLI (below 0.90), and RMSEA (above 0.10). The model unidim2 fits very well the gathered data, with an excellent $\chi^2/df$ (between 2 and 3), and good fits for the CFI, TLI, SRMR and RMSEA (close to 0.05).

Fig 2 shows the item loadings for the unidimensional model (unidim2) that best fits with our gathered data. We can appreciate that eight of all nine item loadings are greater than 0.60, and the loading of item i21 is in 0.50s (greater than the cutoff value of 0.30 for the item loadings). We can also observe that the average of the item loadings is 0.67 with a standard deviation of 0.06 which indicates that no one item significantly affects the variance of flow measure. It means that the DFS-Short BR extracts the variance close equality from all the nine items using the unidimensional model (*unidim1*).

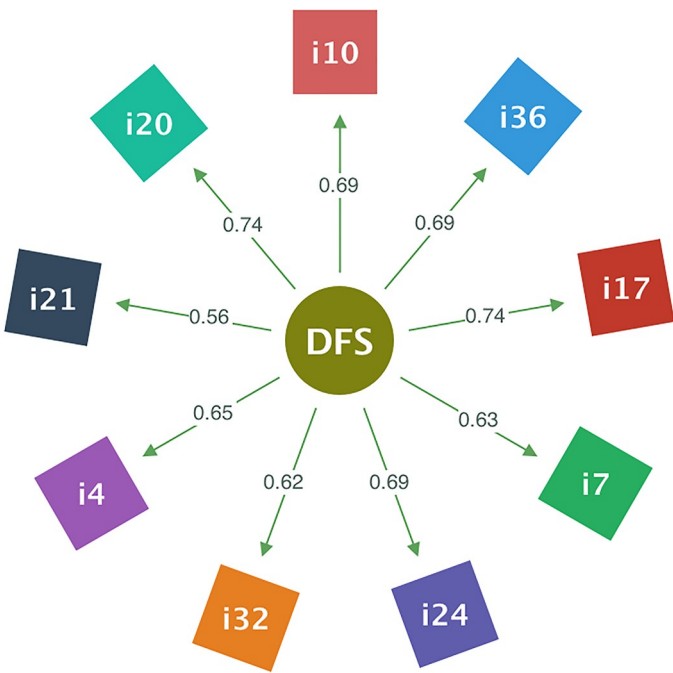

**Fig 2. Item loadings in the unimodel of the DFS-Short BR.**

Regarding the reliability tests, Cronbach's $\alpha$, McDonald's $\omega$ and CR were 0.882. These values indicate good internal consistency for the short version of the DFS-BR using the unidimensional model unidim1.

## Item quality analysis based on IRT

The item quality analysis based on IRT was performed using the Rating Scale Model (RSM), and the Table 7 shows the discrimination and difficulties parameter found for all the 36 items

**Table 7. Discrimination and difficulty item parameters for the DFS-BR.**

|  | a | b1 | b2 | b3 | b4 | bx |
|---|---|---|---|---|---|---|
| i1 | 1.425 | **-3.179** | -2.069 | -0.495 | 1.127 | -1.154 |
| i2 | 1.769 | -2.722 | -1.789 | -0.135 | 1.459 | -0.797 |
| i3 | 0.995 | **-3.088** | -1.358 | 0.784 | 2.926 | -0.184 |
| i4 | 2.09 | -2.599 | -1.531 | -0.313 | 1.209 | -0.809 |
| i5 | 1.944 | -2.623 | -1.381 | -0.029 | 1.496 | -0.634 |
| i6 | 1.344 | -2.311 | -0.777 | 0.532 | 2.048 | -0.127 |
| i7 | 1.933 | -2.623 | -1.577 | -0.124 | 1.521 | -0.701 |
| i8 | 0.97 | -2.440 | -0.718 | 0.842 | 2.248 | -0.017 |
| i9 | 0.728 | **-4.758** | -2.840 | -0.223 | 2.195 | -1.407 |
| i10 | 1.623 | -2.981 | -1.764 | -0.477 | 0.993 | -1.057 |
| i11 | 2.047 | -2.225 | -1.420 | -0.092 | 1.529 | -0.552 |
| i12 | 0.615 | **-4.454** | -1.256 | 1.852 | **5.838** | 0.495 |
| i13 | 0.898 | **-3.273** | -0.989 | 1.237 | **3.799** | 0.194 |
| i14 | 1.938 | -2.988 | -1.936 | -0.713 | 0.673 | -1.241 |
| i15 | 2.268 | -2.760 | -1.408 | -0.108 | 1.360 | -0.729 |
| i16 | 2.029 | -2.169 | -1.087 | 0.149 | 1.522 | -0.396 |
| i17 | 2.849 | -2.340 | -1.108 | 0.138 | 1.397 | -0.478 |
| i18 | 0.758 | -2.666 | -0.432 | 1.482 | **3.167** | 0.388 |
| i19 | 1.057 | **-3.335** | -2.051 | 0.001 | 1.624 | -0.940 |
| i20 | 1.95 | -2.701 | -1.753 | -0.322 | 0.995 | -0.945 |
| i21 | 1.744 | -2.397 | -1.180 | 0.494 | 2.314 | -0.192 |
| i22 | 0.967 | -2.889 | -0.691 | 1.471 | **3.640** | 0.383 |
| i23 | 1.811 | -2.899 | -1.627 | -0.196 | 1.683 | -0.760 |
| i24 | 2.371 | -2.571 | -1.392 | -0.181 | 1.036 | -0.777 |
| i25 | 1.76 | **-3.225** | -1.823 | -0.494 | 1.381 | -1.040 |
| i26 | 2.227 | -2.588 | -1.411 | -0.144 | 1.264 | -0.720 |
| i27 | 2.496 | -2.451 | -1.332 | -0.063 | 1.283 | -0.641 |
| i28 | 0.988 | -2.430 | -0.800 | 0.602 | 2.113 | -0.129 |
| i29 | 0.692 | **-3.650** | -1.569 | 0.888 | **3.013** | -0.330 |
| i30 | 1.54 | **-3.284** | -2.113 | -0.669 | 0.686 | -1.345 |
| i31 | 1.373 | **-3.320** | -1.947 | -0.130 | 1.594 | -0.951 |
| i32 | 1.847 | -2.796 | -1.636 | -0.313 | 1.176 | -0.892 |
| i33 | 1.975 | -2.880 | -1.911 | -0.459 | 1.161 | -1.022 |
| i34 | 0.898 | -2.752 | -1.156 | 0.398 | 1.861 | -0.412 |
| i35 | 0.512 | **-6.082** | **-3.866** | -0.618 | 2.476 | -2.023 |
| i36 | 1.702 | -2.691 | -1.938 | -0.651 | 0.836 | -1.111 |
| M | 1.559 | -2.976 | -1.545 | 0.109 | 1.851 | -0.640 |
| SD | 0.599 | 0.764 | 0.628 | 0.661 | 1.042 | 0.546 |

in the long version of the DFS-BR. The means of item discrimination, parameter (*a*) with $M = 1.599$ and $SD = 0.599$, indicates a high level of discrimination (between 1.35 and 1.69) of our instrument which constitutes an evidence that the DFS-BR has good power to differentiate individual who possess similar level of dispositional flow state. The item with highest discrimination value was the item i17 with value of 2.849 (very-high), and the items with the lowest discrimination value was the item i35 with the value of 0.512 (low discrimination), following by the item i12 with 0.615, and the item i29 with 0.692. The discrimination parameter for the rest of items are above 0.70 which is usually considered the cutoff value in item analysis.

Items with lowest discrimination values (i35, i12 and i29) also show low item loadings in the nine multi-correlated model (see Fig 1) indicating that they may be semantically improved in future versions of the DFS-BR to increase their discrimination power and improve their reliability.

Regarding the difficulty parameters of items (*b1—b4*, and *bx*), when we assessed the response thresholds, no one difficulty average value (*bx*) exceeds the inferior limit of −3 or the superior limit of + 3 defined for the IRT metric. Many items demanding, in the average (*bx*), low levels of the latent factor measure by the DFS-BR (in mean $M = -0.640$ and $SD = 0.546$, easy difficulty level), but the lowest difficulty index was −2.023 that corresponds to the item i35. Lowest difficult levels for psychometric questionnaires indicate that our respondents tend to answer these items with low values. Item difficulty represents the point on the latent trait scale (factor scale measured by item) at which an individual has 50% of endorsing the test item [51, 52].

The item i12 has the average (*bx*) of highest difficulty index with value of 0.495 which is considered a moderate difficulty, far from the limit defined in the IRT metric (+ 3). Some difficult parameters, bold in the Table 7, for the lowest (*b1*) and highest (*b4*) item levels exceed the inferior limit (−3) or the superior limit (+ 3) established in the IRT metric as adequate interval. It means that there were very few responses in the extremes of the 5-point Likert scale for these items. One suggestion for the future versions of the DFS-BR is that these items may be part of studies that improve their meanings, avoiding the discretion of respondents to express their extreme opinions when they answer these items.

For the item analysis of the DFS-Short BR, we depicted the Test Information Curve shown in Fig 3. This curve demonstrates that the largest amount of information measurement is approximately in the range of −4.10 to 2.80, indicating that our instrument is very informative for this range of values. It means that the DFS-Short BR is best suited to measure the dispositional flow state in people who have a latent trait ($\theta$) in this range.

## Discussion

The construct validity and psychometric item quality analysis in an adapted version of the DSF-2 for the Brazilian-Portugues respondents and for general activities was carried out by following the guidelines of the International Test Commission [27]. This process and the results of this validation has been reported in this paper. Our translated items following the backward- and forward procedure have been proved to be acceptable for measuring the nine psychological concepts in which the flow theory is grounded.

The nine multi-correlated model has the best fit for our gathered empirical data in comparison with the second order model (see Table 2). The factor loading of this model was also adequate and all the items competently operate to measure the components for which they were proposed. In the structure validity, we identified that there are no weak correlations for the factor pairs, and that all of them have strong joint variability. The discriminant validity confirmed this observation made in the factor correlations, supporting the assumption of the existence of

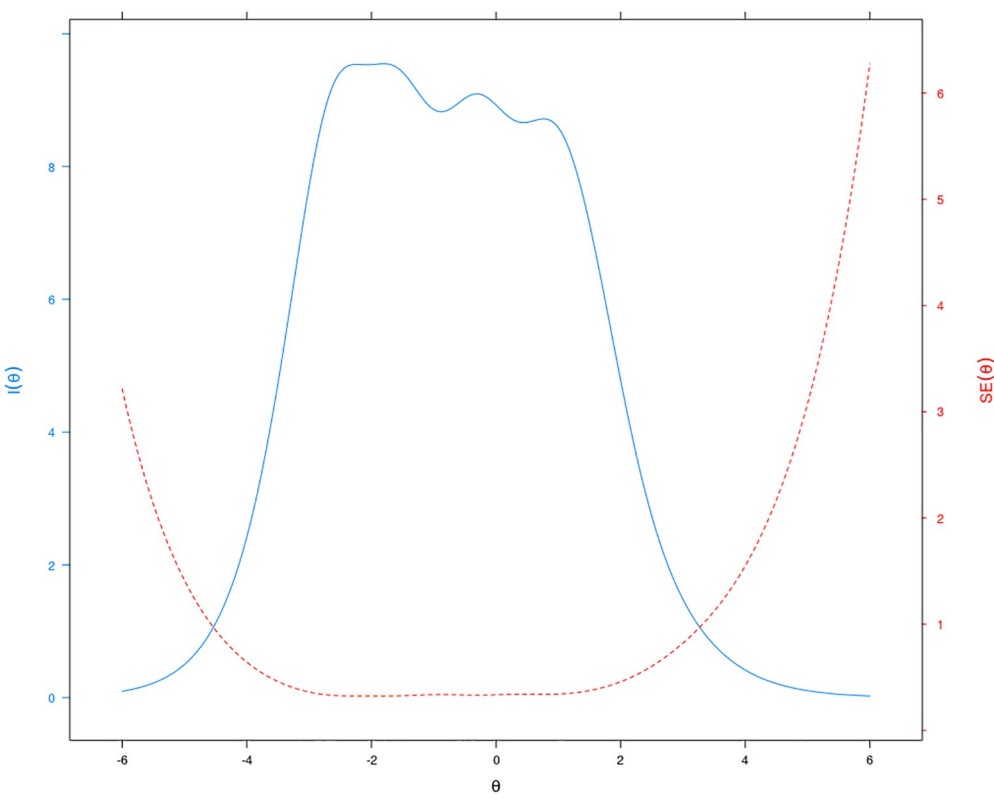

**Fig 3. Test information curve of the DFS-Short BR.**

a psychological concept, known as dispositional flow state, that aggregate the nine factors in only one factor. The results of the convergence validity proved that all of our translated items are adequately related to the factors defined by the flow theory, and in the same manner, in which they were proposed in the original version of the DFS-2. Finally, the reliability tests on the DFS-BR indicated that it presents a good internal consistency for all the nine factors.

In Brazilian, previous studies conducted with adapted versions of the DFS-2 performed construct validity using models with three [19], six [26], and eight [56] factors. The construct validity in our study had nine factors to measure the individuals' tendency towards the flow state. In Portugal, Gouveia et al. (2012) [11] developed a cross-cultural translation of the DFS-2, indicating as the best fit a nine factorial structure, close similar to our study, and with similar item loadings. However, this study, as well as two previous studies conducted in Brazil [24, 25, 57] performed a construct validity with models of nine factors in the domain of physical and exercise activities.

For the general activities, our study and the study conducted by Correia et al. (2020) [26] indicated as an adequate structure a model of nine factors. Although our adapted version of the DFS-2 (DFS-BR) and the adapted version proposed by Correia et al. (2020) have slight differences in the translated items, both instruments are well aligned with the flow theory because their construct validity had been proved adequate for nine factor models. For other languages, several studies have been conducted, and they validate different adaptations of the DFS-2 using nine factors models. Some of them that are highly relevant in the literature, and that have been demonstrated well aligned with the flow theory are the studies conducted for the Italian language [7], for the Japanese language [10], and for the Greek language [58].

In this study, we also proposed and validated a short version of the DFS-2 for the Brazilian-Portuguese language and general activities. This short version known as DFS-Short has been proved with better fit indexes than the nine multi-correlated and the second order model. The reliability tests of this short version also indicated an excellent internal consistency (Chonbach's $\alpha$, McDonald's $\omega$ and CR of 0.80s). Based on these results, we suggest the use of the DFS-Short when a study will be conducted in a context with a time limit, large population to be assessed, and respondents with a busy agenda. In general questionnaires should be limited to 10–15 minutes, and if we considered that, to answer an item, it is needed one min, the necessary time to answer the DFS-BR will be at least 36 minutes, far away from the suggested in [59]. Beside these practical reasons, the good fit indexes found in this study, and the Occam's razor principle (also known as law of parsimony) are motives to defend the idea that the simplest explanation is most likely to be the right one [60, 61], and the DFS-Short BR is the simplest instrument to measure the flow state.

The IRT analysis in this study identified the discrimination and difficulty parameters for all the items of the DFS-BR. These parameters are measurements of item quality, and they were adequate indicating good power of discrimination. The average of difficulty parameters for all the items were in the expected range. The Test Information Curve of the DFS-Short indicated that the latent factor measured by the instrument is very well informative, and that this instrument estimates adequately the individuals' dispositional flow state. In summary, the results presented in this study reveal that our instrument is good enough to be applied in Brazil, and for conducting cross-cultural research studies. Reliability estimation is a fixed value in classical methods, such as the Cronbach's $\alpha$ and McDonald's $\omega$, but in the IRT analysis, this reliability is given by the scale and measurement precision at the item levels [47]. These precision is established by the parameters of difficulty and discrimination for each one of the items.

## Conclusion and future work

In general terms, the long version of the DSF-2 for general activities is an instrument little explored in the Brazilian context, being the study conducted by Correia et al. (in 2020) [26], one of the pioneers to adapt this version and present evidence of validity. In this study, we also conducted the adaptation of the DFS-2 for the Brazilian-Portuguese and validated this instrument, known as DFS-BR, in its long version (with 36 items) and short version (09 items). Our both instruments have been proved to be adequate instruments to measure the individuals' dispositional tendency to flow state as a personality trait. The correlated factorial structure for the long version and its item loadings suggests that each latent factor represents very well the component of the flow theory. However, the reliabilities in many of these factors are low (values of 0.06) which may be indication of the presence of internal structures that should be investigated in future studies. Finally, the power of discrimination and difficulty item estimates for the long and short versions indicate that we are able to accurately differentiate people with similar level of latent traits.

Nowadays, for the Brazilian cultural context, and for the general activities, there have not yet been proposed and validated a short version of the DFS-2. In this paper, we also conducted a study to propose and validate a short version of the DFS-2, known as DFS-Short. This short version constitutes our main contribution to the research literature, and the most relevant practical implication is that the use of few items is more desirable than the use of many items that may require a lot of time, and be tedious for the respondents, as was pointed out by Ziegler et al. (2014) [62].

Another practical implication of our study's findings is that, now, we have two validated instruments (the DFS-BR and DFS-Short) to measure the dispositional flow state in Brazilian

population. These both instruments can not be used for diagnosis, but they can be applied in any Brazilian context and for any activity, to have a better understanding about the engagement phenomena based on the flow theory.

Validation processes of psychometric instruments have two main goals: (1) the first regarding the appropriation and relevance of original concepts in the new culture, and for different domains; (2) the second regarding the validation of the instruments for cross-cultural studies. Although the focus of our study was the former, we also contribute with the latter goal. Many of our translated items were slightly different than the items proposed by Correia et al. (2020) [26]. For instance, the original item i25: *I am not concerned with how I am presenting myself* was translated by Correia et al. (2020) as *Não me preocupo em como me apresento* meanwhile our translation was *Não estou preocupado com a forma como estou me apresentando*. Thus, comparing both validation studies and others that we will conduct in the future for the general activities, through a secondary study of factorial invariance, we can identify problems in the translating idiomatic expressions.

The major limitations in the study reported in this article were the use of a non-probabilistic and non-representative sample. Although the DFS-BR and DFS-Short have been validate as an adequate instrument to measure the individuals' dispositional to flow state, we need to carry out future studies to evaluate how the different groups of Brazilian populations (segment by gender, age, social status and ethnicity) understand the items of the both instruments. Our current dataset is not enough representative and with sufficient sample size to perform this analysis. Thereby, for future studies, we will perform Multi-Group Confirmatory Factor Analysis (MGCF), and also, Differential Item Functioning (DIF) analysis using probabilistic sampling with a representative and bigger sample size that will contains participants from the five regions of Brazil, with different ages, gender, ethnicity and social status.

Criterion validity, also known as external validity of a psychometric instrument, was not performed for the DFS-BR. This kind of validation examines the extent to which scores on an inventory or scale correlate with other external, or non-test criteria. We expected to conduct future studies for performing this kind of validation by evaluating the correlation of flow state with other psychological concepts, such as, anxiety, happiness, serenity, and contentment.

Validation and psychometric properties of self-reported questionnaires depend on the samples, when it has been applied, and its use [63, 64]. In this sense, the evidence presented in this article is not definitive. Along the time, we expect to obtain different validity evidences, in which we will present more information about the strengths, weaknesses, and characteristics of the DFS-BR, in its long and short versions.

In further tests and validations of the DFS-BR, we will conduct studies with focus in the interpretation of the scale values for different groups or subjects. The interpretation of scale values depends also the context, and although we presented here validation for general activities, each response has a specific activity. These activities and others (that will be gathered in the future) will define the context in which we will analyze and interpret the gathered values for the dispositional flow scale. Establishing cut-off points to define what mean low or higher values of dispositional flow scale is another relevant future study that we expect to conduct in the future. This kind of study is also an important issue in the adaptation and construction of psychometric instrument that also depends on the population and context in which it is applied.

## Supporting information

**S1 Appendix. DFS-BR.** Portuguese-Brazilian Long Version of the Dispositional Flow Scale 2. (PDF)

**S2 Appendix. DFS-Short BR.** Portuguese-Brazilian Short Version of the Dispositional Flow Scale 2.
(PDF)

**S1 Dataset. DFS Dataset.** Dataset with the responses gathered to validate the DFS-BR.
(CSV)

## Author Contributions

**Conceptualization:** Ig Ibert Bittencourt, Yu Lu, Sheyla Fernandes, Jorge Coelho, Seiji Isotani.

**Data curation:** Ig Ibert Bittencourt, Leogildo Freires, Geiser Chalco Challco, Júlio Costa.

**Funding acquisition:** Ig Ibert Bittencourt, Yu Lu.

**Investigation:** Ig Ibert Bittencourt.

**Methodology:** Ig Ibert Bittencourt, Geiser Chalco Challco, Sheyla Fernandes, Jorge Coelho, Alexandre Marinho, Seiji Isotani.

**Project administration:** Ig Ibert Bittencourt, Yu Lu, Alexandre Marinho.

**Resources:** Alexandre Marinho.

**Software:** Alexandre Marinho.

**Supervision:** Ig Ibert Bittencourt, Seiji Isotani.

**Validation:** Ig Ibert Bittencourt, Leogildo Freires, Geiser Chalco Challco, Sheyla Fernandes, Jorge Coelho, Júlio Costa.

**Visualization:** Ig Ibert Bittencourt, Geiser Chalco Challco, Jorge Coelho, Alexandre Marinho.

**Writing – original draft:** Ig Ibert Bittencourt, Leogildo Freires, Geiser Chalco Challco, Sheyla Fernandes.

**Writing – review & editing:** Ig Ibert Bittencourt, Yu Lu, Geiser Chalco Challco, Yang Pian, Seiji Isotani.

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
