## [Decision Letter · Decision Letter 0]

5 Feb 2021

PONE-D-20-30707

Validation and psychometric properties of the Brazilian-Portuguese Dispositional Flow Scale 2 (DFS-BR)

PLOS ONE

Dear Dr. Bittencourt

Thank you for submitting your manuscript to PLOS ONE. After careful consideration, we feel that it has merit but does not fully meet PLOS ONE’s publication criteria as it currently stands. Therefore, we invite you to submit a revised version of the manuscript that addresses the points raised during the review process.

We look forward to receiving your revised manuscript.

Kind regards,

Paolo Roma

Academic Editor

PLOS ONE

Journal Requirements:

2.We note that you have indicated that data from this study are available upon request. PLOS only allows data to be available upon request if there are legal or ethical restrictions on sharing data publicly. For more information on unacceptable data access restrictions, please see http://journals.plos.org/plosone/s/data-availability#loc-unacceptable-data-access-restrictions.

3.We note that the grant information you provided in the ‘Funding Information’ and ‘Financial Disclosure’ sections do not match.

Additional Editor Comments :

Dear Authors,

as already mentioned, it is of paramount importance to stress out the differences between this study and the already existing validation in brazilian-portuguese of the Dispositional Flow Scale 2.

Furthermore, as highlighted by reviewer1, the manuscript requires major revisions.

Reviewers' comments:

Reviewer's Responses to Questions

**Comments to the Author**

1. Is the manuscript technically sound, and do the data support the conclusions?

Reviewer #1: Yes

Reviewer #2: Yes

2. Has the statistical analysis been performed appropriately and rigorously? 

Reviewer #1: Yes

Reviewer #2: Yes

3. Have the authors made all data underlying the findings in their manuscript fully available?

Reviewer #1: Yes

Reviewer #2: Yes

4. Is the manuscript presented in an intelligible fashion and written in standard English?

Reviewer #1: No

Reviewer #2: Yes

5. Review Comments to the Author

Reviewer #1: General comments

The study at hand aimed to validate and assess the psychometric properties of a widely adopted questionnaire for estimating dispositional flow in a variety of activities. The authors have done a great job in gathering and analyzing the data, providing satisfactory results for supporting the scale’s psychometric properties. However, despite the quality of data analysis, there are still major concerns undermining a favorable recommendation for its publication.

The introduction is mostly descriptive, it does not highlight a research gap or provide reasoning behind the decision to conduct such a study and it does not offer possible practical applications for the instrument. As the study pertains to the Brazilian context, at least a brief literature review of flow studies in Brazil is deemed necessary, as a variety of flow studies have been conducted through the use of other instruments or even translated versions of this same scale (DFS-2). For instance, a doctorate thesis by Simone Salvador Gomes (2014) has also assessed the DFS-2 psychometric properties in Brazil, which was used in other publications; and a masters’ dissertation, by Cássia Roettgers, has developed and validated another flow scale for Brazil.

Meanwhile, the Dispositional Flow Scale (DFS-2) has already been validated for the Portuguese language by Gouveia, Ribeiro, Marques & Carvalho (2012) in Portugal and has recently been validated for Brazilian Portuguese by Correia, Mendonça Filho, Tischer, Oliveira & Giacomoni (2020, doi: 10.1007/s43076-020-00028-0) and was not acknowledged or discussed in the present study. Why is another validation necessary? Moreover, validation studies for other countries were not discussed as well.

Regarding the methods section, sample characteristics were only briefly described. As flow is experienced during an activity and is also studied in regards to specific contexts such as work, study or sports & exercise, more information is needed in order to fully describe the sample. What activities do these subjects practice and for how long? What were the inclusion/exclusion criteria? Were specific Facebook groups targeted for recruitment, such as sports or music groups? Did the authors use a sociodemographic questionnaire to assess sample characteristics? What regions of Brazil are being represented? I also suggest presenting the sample’s flow scores for all dimensions in the long and short versions, to better characterize the subjects.

Another method limitation was the lack of other measurements, such as other variables that would be expected to correlate with flow, thus, no external evidences of validity were presented. I also suggest assessing the scale’s Average Variance Extracted.

For the CFA results, did the model reach satisfactory fit in the first try or were there adjustments made based on modification indices or any other criteria? As multi-group analysis was suggested for future work, why not include measurement invariance tests for the present study?

Major improvements are also required at the discussion section, which is very brief and carries the same limitations as the introduction. Considering that most of the readers might not be familiar with Item Response Theory, what does it add to the validation process and how can its results be applied? The study limitations and practical applications also need to be described.

For the data set provided as supporting material, subjects’ full name are being disclosed and need to be coded (for example, subject A, B, C…) or hidden. The csv file has also merged the information from all columns into only two, making it difficult to be analyzed.

Finally, language must be reviewed as there are many flaws throughout the text.

In this sense, a major review is necessary in order to better justify the present work and situate it within the literature, especially considering that a Brazilian-Portuguese DFS-2 validation study has already been published in 2020.

Reviewer #2: Reviewer Comments for Manuscript PONE-D-20-30707: Validation and psychometric properties of the Brazilian-Portuguese Dispositional Flow Scale 2 (DFS-BR)

General Remarks:

I read this manuscript with great interest. Overall, the manuscript addresses an interesting and valuable issue. It also provides a measurement tool in the field of sports psychology. The study validated the psychometric properties of an adapted version of the DFS-2 for the Brazilian Portuguese language.

The study has a number of strengths. First, the information presented in this manuscript is consistent with the journal’s mission. The manuscript is clearly within the journal’s mission, and there is a relevance to the readership. Second, the study sample was chosen appropriately and described in sufficient detail for the results to be generalized. Finally, the data were analyzed with correct analysis methods and the findings obtained from the analysis are very strong.

6. PLOS authors have the option to publish the peer review history of their article (what does this mean?). If published, this will include your full peer review and any attached files.

Reviewer #1: No

Reviewer #2: **Yes: **Gözde Ersöz

---

## [Author Response · Author response to Decision Letter 0]

5 Mar 2021

Dear reviewers and editors of PLOS ONE, we would like to thank for your insightful comments and suggestions; each one has been carefully analyzed and considered in the new version of the paper. 

To facilitate and organize our changes in the manuscript, we cluster reviewers’ comments that have similar suggestions. We also included in this letter parts of the text that were changed or included in the manuscript to illustrate how we address each comment. 

We did our best to answer all comments thoroughly. We hope that all these changes fulfill the requirements to make the manuscript acceptable for its publication.

Looking forward to hearing from you.

Comment 1 – Editor: It is of paramount importance to stress out the differences between this study and the already existing validation in brazilian-portuguese of the DFS 2.

Answer: Dear editor, thank you for your time and review. To make clear and explain in a better way what are the differences between our study and the study of Correia et al. (2020), we made the following changes in the article. Please, see below.

• In Section “Introduction,” we changed the 6th paragraph (lines 53 to 62), indicating the lack of validation of the short version of DFS-2 in the study of Correia et al. (2020), and we also indicated the use of IRT in our study as complement to the classical validation carried out by Correia et al. (2020).

Only, a recent study conducted by Correia et al. (2020) [26] performed the validation of the DFS-2 in the Brazilian-Portuguese language for general activities. However, this study did not perform validation of the short version of DFS-2, and only classical validation methods (construct validity and internal consistency) were carried out. For these reasons, and to complement results from previous validation studies, we conducted a study reported in this paper in which, in addition to the classical validation methods, we performed a psychometric item quality analysis of the instrument using Item-Response Theory (IRT). We also conducted a validation of the short-version of the DFS-2 for the Portuguese-Brazil language.

• In Section “Discussion,” we added the 4th paragraph (lines 430 to 439) to briefly describe the similarities of our obtained results and the study of Correia et al. (2020), as well as other validation studies conducted in other languages.

Only, a recent study conducted by Correia et al. (2020) [26] performed the validation of the DFS-2 in the Brazilian-Portuguese language for general activities. However, this study did not perform validation of the short version of DFS-2, and only classical validation methods (construct validity and internal consistency) were carried out. For these reasons, and to complement results from previous validation studies, we conducted a study reported in this paper in which, in addition to the classical validation methods, we performed a psychometric item quality analysis of the instrument using Item-Response Theory (IRT). We also conducted a validation of the short-version of the DFS-2 for the Portuguese-Brazil language.

• In Section “Conclusion and future work,” in the 1st paragraph (lines 467 to 472), we highlighted again the difference between our study and Correia et al. (2020) as follows:

In general terms, the long version of the DSF-2 for general activities is an instrument little explored in the Brazilian context, being the study conducted by Correia et al. (in 2020) [26], one of the pioneers to adapt this version and present evidence of validity. In this study, we also conducted the adaptation of the DFS-2 for the Brazilian-Portuguese and validated this instrument, known as DFS-BR, in its long version (with 36 items) and short version (09 items). 

Comment 1 – Review 1: The introduction is mostly descriptive; it does not highlight a research gap or provide reasoning behind the decision to conduct such a study and it does not offer possible practical applications for the instrument. 

Comment 8 – Review 1: Major improvements are also required at the discussion section, which is very brief and carries the same limitations as the introduction.

Answer: Dear reviewer, thank you for the review. We have rewritten and added in the article some parts to highlight the relevance of our study as follows: 

• In Section “Introduction,” we added the 5th paragraph (lines 37 to 50) to highlight the importance to validate more than one translated version of a psychometric instrument, and what practical implication has it.

Despite the possibility to buy and obtain a translated version of the DFS-2 for the Portuguese-Brazil language (from the website mindgarden.com). Few published studies have been conducted in Brazil to demonstrate its validity. When a psychometric instrument is translated to a new language and introduced in a new cultural context, for its validation, ensuring its conceptual and idiomatic equivalence is necessary [16, 17]. Translation of items needs to maintain the same equivalence in what are measured and operationalized in its original language, and the adaptation for a new culture should consider the relevance of concepts and the domains in which the instrument is applied [18]. Thus, it is necessary to evaluate the appropriateness of each item in terms of the ability of respondents to represent the concepts measured by the instruments in different domains. In this sense, various validation studies should be conducted in the new culture, in different domains (including, general activities), with different populations, and using different versions of the same instruments (this last for cross-cultural validation).

• In Section “Introduction,” we updated the 6th paragraph (lines 51 to 60), complementing the previous studies conducted in Brazil by indicating the domain in which they were applied. We highlight the lack of studies for general activities, and briefly presented the differences of our study in comparison with the recent study conducted by Correia et al. (2020) for general activities.

In Brazil, studies to validate the Portuguese-Brazilian version of the DFS-2 were conducted only in few specific domains, such as the work of Freitas et al. (2019) [19], and the sport and physical activities [20, 21, 22, 23, 24, 25]. Only, a recent study conducted by Correia et al. (2020) [26] performed the validation of the DFS-2 in the Brazilian-portuguese language for general activities. However, this study did not perform validation of the short version of DFS-2, and only classical validation methods (construct validity and internal consistency) were carried out. For these reasons, and to complement results from previous validation studies, we conducted a study reported in this paper in which, in addition to the classical validation methods, we performed a psychometric item quality analysis of the instrument using Item-Response Theory (IRT). We also conducted a validation of the short-version of the DFS-2 for the Portuguese-Brazil language.

• In Section “Discussion,” in the 5th paragraph (lines 444-453), where we summarized the validation results on our short version of the DFS-2 (DFS-Short), we highlighted the practical implications of having a validated short version of the DFS-2.

Based on these results, we suggest the use of the DFS-Short when a study will be conducted in a context with a time limit, large population to be assessed, and respondents with a busy agenda. In general questionnaires should be limited to 10-15 mins, and if we considered that, to answer an item, it is needed one min, the necessary time to answer the DFS-BR will be at least 36 mins, far away from the suggested in [56]. Beside these practical reasons, the good fit indexes found in this study, and the Occam’s razor principle (also known as law of parsimony) are motives to defend the idea that the simplest explanation is most likely to be the right one [57, 58], and the DFS-Short BR is the simplest instrument to measure the flow state.

• In Section “Conclusion and future work,” we highlighted the main contrition of our study (the validation of short version of DFS-2) by adding the 2nd paragraph (lines 479-485).

Nowadays, for the Brazilian cultural context, and for the general activities, there have not yet been proposed and validated a short version of the DFS-2. In this paper, we also conducted a study to propose and validate a short version of the DFS-2, known as DFS-Short. This short version constitutes our main contribution to the research literature, and the most relevant practical implication is that the use of few items is more desirable than the use of many items that may require a lot of time, and be tedious for the respondents, as was pointed out by Ziegler et al. (2014) [59].

• In Section “Conclusion and future work,” we added the 3rd paragraph (lines 479-485) to indicate the limits in the practical application of our validated instruments.

Another practical implication of our study’s findings is that now we have two validated instruments (the DFS-BR and DFS-Short) to measure the dispositional flow state in Brazilian population. These both instruments cannot be used for diagnosis, but they can be applied in any Brazilian context and for any activity, to have a better understanding about the engagement phenomena based on the flow theory.

• Finally, in Section “Conclusion and future work,” we added the 4th parameter (lines 491-501) to briefly present the contribution of our study for the psychometrics in positive psychology. We highlighted the importance of our obtained findings in comparison with the recent study conducted by Correia el al. (2020).

Validation processes of psychometric instruments have two main goals: (1) the first regarding the appropriation and relevance of original concepts in the new culture, and for different domains; (2) the second regarding the validation of the instruments for cross-cultural studies. Although the focus of our study was the former, we also contribute with the latter goal. Many of our translated items were slightly different than the items proposed by Correia et al. (2020) [26]. For instance, the original item i25: I am not concerned with how I am presenting myself was translated by Correia et al. (2020) as Não me preocupo em como me apresento meanwhile our translation was Não estou preocupado com a forma como estou me apresentando. Thus, comparing both validation studies and others that we will conduct in the future for the general activities, through a secondary study of factorial invariance, we can identify problems in the translating idiomatic expressions.

Comment 2 – Review 1: As the study pertains to the Brazilian context, at least a brief literature review of flow studies in Brazil is deemed necessary, as a variety of flow studies have been conducted through the use of other instruments or even translated versions of this same scale (DFS-2). For instance, a doctorate thesis by Simone Salvador Gomes (2014) has also assessed the DFS-2 psychometric properties in Brazil, which was used in other publications; and a masters’ dissertation, by Cássia Roettgers, has developed and validated another flow scale for Brazil. Meanwhile, the Dispositional Flow Scale (DFS-2) has already been validated for the Portuguese language by Gouveia, Ribeiro, Marques & Carvalho (2012) in Portugal and has recently been validated for Brazilian Portuguese by Correia, Mendonça Filho, Tischer, Oliveira & Giacomoni (2020, doi: 10.1007/s43076-020-00028-0) and was not acknowledged or discussed in the present study. Why is another validation necessary? Moreover, validation studies for other countries were not discussed as well.

Answer: Dear reviewer, thank you for your comment. We added citation to the research studies that you indicated, as well as other studies conducted in the Brazil.

• In Section “Introduction,” 6th paragraph (lines 51 to 55), we briefly described the domains in which the previous Brazilian studies were conducted as follows:

In Brazil, studies to validate the Portuguese-Brazilian version of the DFS-2 were conducted only in few specific domains, such as the work of Freitas et al. (2019) [19], and the sport and physical activities [20, 21, 22, 23, 24, 25]. Only, a recent study conducted by Correia et al. (2020) [26] performed the validation of the DFS-2 in the Brazilian-portuguese language for general activities. However, this study did not perform validation of the short version of DFS-2, and only classical validation methods (construct validity and internal consistency) were carried out.

• In Section “Discussion,” 3rd paragraph (lines 422 to 429), we briefly summarized the differences and similarities of our findings, and findings from previous Brazilian studies.

In Brazilian, previous studies conducted with adapted versions of the DFS-2 performed construct validity using models with three [19], six [26], and eight [53] factors. The construct validity in our study had nine factors to measure the individuals’ tendency towards the flow state. In Portugal, Gouveia et al. (2012) [11] developed a cross-cultural translation of the DFS-2, indicating as the best fit a nine-factorial structure, close similar to our study, and with similar item loadings. However, this study, as well as two previous studies conducted in Brazil [54, 24, 25] performed a construct validity with models of nine factors in the domain of physical and exercise activities.

• In Section “Discussion,” in the 4th paragraph (lines 430-439), we cited studies conducted in other countries and with other language, but that achieved same finding in the structural validation.

For the general activities, our study and the study conducted by Correia et al. (2020) [26] indicated as an adequate structure a model of nine factors. Although our adapted version of the DFS-2 (DFS-BR) and the adapted version proposed by Correia et al. (2020) have slight differences in the translated items, both instruments are well aligned with the flow theory because their construct validity had been proved adequate for nine factor models. For other languages, several studies have been conducted, and they validate different adaptations of the DFS-2 using nine-factors models. Some of them that are highly relevant in the literature, and that have been demonstrated well aligned with the flow theory are the studies conducted for the Italian language [7], for the Japanese language [10], and for the Greek language [55].

Comment 3 – Review 1: Regarding the methods section, sample characteristics were only briefly described. As flow is experienced during an activity and is also studied in regard to specific contexts such as work, study or sports & exercise, more information is needed in order to fully describe the sample. What activities do these subjects practice and for how long? What were the inclusion/exclusion criteria? Were specific Facebook groups targeted for recruitment, such as sports or music groups? Did the authors use a sociodemographic questionnaire to assess sample characteristics? Were specific Facebook groups targeted for recruitment, such as sports or music groups? What regions of Brazil are being represented? I also suggest presenting the sample’s flow scores for all dimensions in the long and short versions, to better characterize the subjects.

Answer: Dear reviewer, thank you for your comment. We rewritten the section “Participants” (lines 64 to 89) to a provided more detailed information of our sample, our sociodemographic questionnaires, and activities assessed by the DFS-2.

We employed a non-probabilistic sample (by voluntary sampling) of n = 681 Brazilians as participants in this study. All of them, from 18 to 74 years old, with an average age of M = 27.17 years old, and SD = 12.29.

A self-reported socio-demographic questionnaire was answered by the participants to gather information of their gender, civil status, ethnicity, socio-economic status, and sexual orientation. The majority of participants were men 59.03%, 40.09% were female, and 0.88% did not declare their gender. The civil status of 70.63% participants was single, 20.12% were married, 6.75% were cohabiting with a partner, 2.06% were divorced, and 0.44% were widow. Regarding their ethnicity, 61.81% declared to be white, 23.20% declared to be pardo (mixed-race), 7.64% declared to be black, 2.06% declared to be yellow (mongoloid asian), 0.59% declared to be part of indigenous tribal population, and 4.70% preferred not to declare their ethnicity. Social economic status of participants were: 46.70% of the middle class, 32.89% of middle lower class, 9.69% of lower class, 8.81% of middle upper class, and 1.91% of higher economic class. Sexual orientation of participants: 83.26% declared themselves as heterosexual/straight, 5.29% declared to be bi-sexual, 4.11% declared to be homosexual, 0.15% declared to be transsexual, and 7.20% did not wish to declare their sexual orientation.

In order to be participants of this study, respondents needed to declare themselves as Brazilian citizens, residing in Brazil, and having fluency in Portuguese language. Inconsistent answers were removed from the data, and the respondents of these answers were not considered part of participants.

In the questionnaire used to gather the data, respondents were also indicated the activity for which the dispositional flow state was measured. These activities were: learning about physics (33.14%), security information (24.93%), and other 192 activities (41.93%, with less than 3% per activity).

Answer: No one specific groups were targeted in our study. Thereby, we added this sentence in section “Recruitment” (lines 94 and 95).

We performed the data collection entirely through the Internet, using volunteer participation, in which the involved researchers sent recruitment messages by their own social media networks (e.g., Facebook, Instagram, Whatsapp) and email to obtain responses to the questionnaires used in this study. No one specific groups were targeted in the social media networks. 

Answer: We included a statistic descriptive of data set employed in the study in Table 1 for the section “Data analysis procedure” (lines 152 to 153). We decided don’t include the sample’s flow scores for all dimensions in the long and short versions because we don’t have the purpose to make comparison of flow scores for different population, and we also don’t have the goal to obtain a validated instrument to diagnosing. We intend to validate a psychometric instrument to be used in empirical studies and cross-cultural studies to understand in better way the flow (engagement) phenomenon.

Comment 4 – Review 1: Another method limitation was the lack of other measurements, such as other variables that would be expected to correlate with flow, thus, no external evidences of validity were presented.

Answer: Dear reviewer, Your observation its highly relevant. One limitation of our study was the lack of external validity. We forgot mention it in the first submitted version of paper. For the new version, we clearly indicated this limitation in the Section “Conclusion and future work” (lines 514-519).

Criterion validity, also known as external validity of a psychometric instrument, was not performed for the DFS-BR. This kind of validation examines the extent to which scores on an inventory or scale correlate with other external, or non-test criteria. We expected to conduct future studies for performing this kind of validation by evaluating the correlation of flow state with other psychological concepts defined in the positive psychology as equivalent to well-being, such as, happiness, serenity, and contentment.

Comment 5 – Review 1: I also suggest assessing the scale’s Average Variance Extracted.

Answer: Dear reviewer, Thank you for the suggestion. AVE is part of convergent and discriminant validity. An important validation process that we didn’t included in the previous version of our article. For the new version, we decided to add a complete section “Convergent and discriminant validity” (lines 299-336) in which we detailed the results of calculated AVE, composite reliability (CR) and Heterotrait-monotrait (HTMT) ratio to assess the discriminant and convergence of the DFS-BR. This section is detailed as follow.

Table 5 shows the CR, AVE, VIF, square root of AVEs, factor correlations, and HTMT ratios. As we explained in the data analysis procedure, these values are indexes used to evaluate the convergent and discriminant validity of the DFS-BR. Convergent validity is assessed by the item loadings, as well as the CR and AVE values. All the AVE values are less than 0.50, the cutoff frequently defined to ensure convergence. Four of nine factors (UF, SC, TT, AE) have at least one item with high level of convergence (item loadings greater than 0.70), and the CR values indicate acceptable (0.60s) and good (0.70s) convergence for all the factors. Based on these results, we state that the multi-correlation model has an acceptable convergence validity. This statement is based on the structure equation modelling literature in which it is said that the AVE is a very conservative test [52]. A convergent validity is still acceptable when the AVE values are less than the cutoff of 0.50, but all the CR values are greater than 0.6 [38].

Based on Table 5, we can observe that there are no multicollinearity problems because all the VIF values are lower than 5 (cutoff value). If VIF values are greater than 5, and the correlations are greater than 0.80, it is suggested the combination of the factors [51]. However, this suggestion is subject to the underpinning theory. No one VIF value is greater than 5, so that there is no need to combine any factor in the multi-correlated model based on this criterion.

To assess the discrimination validity of the multi-correlated model based on the Fornell-Larcker criterion [38], and the similarity degree between factors [39], the columns to the right of VIF values are composed by the factor correlations (lower triangular part), their HTMT ratio (upper triangular part), and the square root of AVE values (diagonal bold values). According to the Fornell-Larcker criterion, when the square root of the AVE value of each factor is greater than its correlations, it is an indication that the difference between each measurement factor is better [51], demonstrating a good discriminant validity. Table 5 shows that this criterion has not been satisfied for the multi-correlated model. The upper triangular part of this Table shows the HTMT ratios of correlations. With exception of the pairs CG-CTH and CG-SC, all the rest of correlation pairs have HTMT ratios greater than 0.85 indicating discriminant problems for the six factors of Challenge-Skill Balance (CSB), Merging of Action-Awareness (MAA), Unambiguous Feedback (UF), Loss of Self-Consciousness (LSC), Transformation of Time (TT), and Autotelic Experience (AE). Although the lack of discriminant validity can be understood as overlapping items in the factors of a psychometric instrument, or that the factors are measuring the same thing, in our case, we consider these high HTMT values as a confirmation that the nine factors can be aggregated in only one psychological concept, which is known as the individual’s dispositional tendency towards flow state according to the Csikszentmihalyi flow theory [1].

Comment 6 – Review 1: For the CFA results, did the model reach satisfactory fit in the first try or were there adjustments made based on modification indices or any other criteria?

Answer: In Section “Data analysis procedure,” we complement the 3rd paragraph (lines 161-164), answering your question as detailed as follows.

... There was no need to establish fixing parameters, starting values, modifiers, or error values in the models. We only fixed to 1 the interfactor correlations of factors with values greater than 1 because WLSMV yielded to give a moderate overestimation for this parameter [31].

Comment 7 – Review 1: As multi-group analysis was suggested for future work, why not include measurement invariance tests for the present study?

Answer: Dear reviewer, we don’t have a representative sample and enough sample size to conduct the invariance tests. Therefore, we included this limitation in the Section “Conclusion and future work,” we added the 5th paragraph (lines 503-513) in which we mentioned and clarify this limitation and also highlight the importance to conduct the MGCFA in future studies.

The major limitations in the study reported in this article were the use of a non-probabilistic and non-representative sample. Although the DFS-BR and DFS-Short have been validate as an adequate instrument to measure the individuals’ dispositional to flow state, we need to carry out future studies to evaluate how the different groups of Brazilian populations (segment by gender, age, social status and ethnicity) understand the items of both instruments. Our current dataset is not enough representative and with sufficient sample size to perform this analysis. Thereby, for future studies, we will perform Multi-Group Confirmatory Factor Analysis (MGCF), and also, Differential Item Functioning (DIF) analysis using probabilistic sampling with a representative and bigger sample size that will contains participants from the five regions of Brazil, with different ages, gender, ethnicity and social status.

Comment 9 – Review 1: Considering that most of the readers might not be familiar with Item Response Theory, what does it add to the validation process and how can its results be applied? The study limitations and practical applications also need to be described.

Answer: Dear reviewer, thank you for your comment. Usually, we forgot that our writings are for a broad target population, and we assumed that any reader clearly understands our applied data-analysis methods. We don’t have a representative sample and enough sample size to conduct the invariance tests. Thus, we included in the section “Data analysis procedure” the 7th paragraph (lines 199-215) in which we briefly detailed the importance of IRT.

For the analysis of the psychometric item quality in both versions of the DFS-BR, we employed the IRT analysis. It is a framework to assess psychometric instruments by explaining the dependencies between item responses within a person and between persons [44]. IRT has been developed to fill gaps of the Classical Test Theory (CTT), such as the impossibility of assessing the individual parameters for each item, and the unrealistic assumption that there is uniformity of confidence intervals for a person’s latent construct [45]. In this study, the dispositional flow state is the latent construct measured by the instrument, and the use of IRT leads us to have a more realistic confidence intervals for this measure wherein there are different levels of item parameters used for different levels of dispositional flow state. These parameters are the difficulty and discrimination of items. The item difficulty, based on the item responses within all participants, determines the manner in which the item behaves along the measured scale, so that this parameter in each item level is an estimate of the dispositional flow state level to pass this level. Item discrimination indicates the endorsing degree for a correct item level given a latent constructor level, so this value determines the quality of item to differentiate similar levels of the latent construct being measured by the psychometric instrument.

Comment 10 – Review 1: For the data set provided as supporting material, subjects’ full name are being disclosed and need to be coded (for example, subject A, B, C…) or hidden. The csv file has also merged the information from all columns into only two, making it difficult to be analyzed.

Answer: Dear reviewer, we fixed the problem in the exported dataset, and also, hidden the sensible information (full names of respondents).

Comment 11 – Review 1: Finally, language must be reviewed as there are many flaws throughout the text.

Answer: Dear reviewers, thank you for pinpointing these issues. We have proofread our paper to address all these comments.

---

## [Decision Letter · Decision Letter 1]

12 Apr 2021

PONE-D-20-30707R1

Validation and psychometric properties of the Brazilian-Portuguese Dispositional Flow Scale 2 (DFS-BR)

PLOS ONE

Dear Dr. Pinto,

Thank you for submitting your manuscript to PLOS ONE. After careful consideration, we feel that it has merit but does not fully meet PLOS ONE’s publication criteria as it currently stands. Therefore, we invite you to submit a revised version of the manuscript that addresses the points raised during the review process.

We look forward to receiving your revised manuscript.

Kind regards,

Paolo Roma

Academic Editor

PLOS ONE

Journal Requirements:

Reviewers' comments:

Reviewer's Responses to Questions

**Comments to the Author**

1. If the authors have adequately addressed your comments raised in a previous round of review and you feel that this manuscript is now acceptable for publication, you may indicate that here to bypass the “Comments to the Author” section, enter your conflict of interest statement in the “Confidential to Editor” section, and submit your "Accept" recommendation.

Reviewer #1: All comments have been addressed

Reviewer #3: (No Response)

2. Is the manuscript technically sound, and do the data support the conclusions?

Reviewer #1: Yes

Reviewer #3: Yes

3. Has the statistical analysis been performed appropriately and rigorously? 

Reviewer #1: Yes

Reviewer #3: Yes

4. Have the authors made all data underlying the findings in their manuscript fully available?

Reviewer #1: Yes

Reviewer #3: Yes

5. Is the manuscript presented in an intelligible fashion and written in standard English?

Reviewer #1: Yes

Reviewer #3: Yes

6. Review Comments to the Author

Reviewer #1: (No Response)

Reviewer #3: The article addresses an interesting question, and in this sense, there are relevant strengths because the manuscript addresses a relevant issue that is the adaptation to Brazilian-portuguese dispositional flow scale 2, that could be relevant to increase the literature about the optimal engagement.

There are some issues that, I believe, could contribute to improve the paper:

1. In general terms, the validation process of a test never really ends, nor could it be said that there is any definitive evidence regarding it. In other words, different validity evidences are obtained along the time (utility, content, construct, appearance, consequences, criteria, ...). For these reasons I would suggest including some ideas and reflexions about it in the conclusions and what could be the next step in order to obtain other validity evidences. On the other hand, in the forthcoming researches, studies about the interpretation of the score will be welcome and the study of cutoff points in the scale. This important issue in the adaptation and construction of a measurement instrument should be comment.

2. There were missing data? If yes, how they were treated (deleted, multiple imputation method,…).

3. In the descriptive statistics the skewness and kurtosis should be reported in order to justify the estimation method used.

4. If the authors would like report an index to justify the adequacy of the sample size, in CFA there is one specific global fit index for that purpose (Critical N). Given the statistical procedure follow, the lines 147 to 153 could be deleted.

5. As the authors do, the polychoric correlations should be used and, at least, a description of its adequacy should be commented. A good reference to do that is:

Holgado-Tello, F.P., Carrasco, M.A., Barrio, Mª.V., y Chacón, S. (2009). Factor Analysis of the Big Five Questionnaire using polychoric correlations in children. Quality & quantity, the International Journal of Methodology, 43(1), 75-85.

If it is possible, the Bonferroni correction for the tests of adequacy of polychoric correlations should be reported.

6. The procedure followed to shorten the scale presents problems that should not be avoided. In this method, only reliability information is taken into account, and it has as a negative consequence that, if the reliability and discriminations is optimized, the validity suffers a deterioration. That is, the reliability increases at the cost of validity. This issue should be mentioned.

7. The adequacy of the model should be based on the global fit indexes. In this sense, if there are individual parameters that has not statistical relevance, but however, the global fit indexes are adequate, and they are conceptual congruence, there are not arguments to delete they from the model. Sometimes, the redaction in this sense is very restrictive. I could recommend taken into account, also, the sense of the items, and the global fit indices. In CFA perspective, the fit indexes should be related to the global of the model. In this sense, if the model fit, although there are individual parameters close to 0, or with low values, they could be considered relevant if they have coherence and congruence with the theoretical assumptions.

8. GFI should be reported.

9. In general terms, the reliabilities of the scales are very low, this could be indicate that alternatives internal structures should be investigate.

In sum, the paper is well written, and it could contribute to the accumulation of empirical validity evidences about DSF scale. In this sense is an interesting paper that could increase the literature with relevant results.

In conclusion, I would like to emphasize that I understand that the vision of the reviewers, at times, can seem idiosyncratic, and from this position I would not like to be dogmatic or rigid in my approaches. I have only tried to offer some suggestions on how the article could be improved, which may or may not be considered.

7. PLOS authors have the option to publish the peer review history of their article (what does this mean?). If published, this will include your full peer review and any attached files.

Reviewer #1: **Yes: **Renan Codonhato

Reviewer #3: No

---

## [Author Response · Author response to Decision Letter 1]

21 Apr 2021

Comment 1 – Reviewer 3: There are some issues that, I believe, could contribute to improve the paper: ... 1. In general terms, the validation process of a test never really ends, nor could it be said that there is any definitive evidence regarding it. In other words, different validity evidences are obtained along the time (utility, content, construct, appearance, consequences, criteria, ...). For these reasons I would suggest including some ideas and reflexions about it in the conclusions and what could be the next step in order to obtain other validity evidences. On the other hand, in the forthcoming researches, studies about the interpretation of the score will be welcome and the study of cutoff points in the scale. This important issue in the adaptation and construction of a measurement instrument should be comment.

Answer: Dear reviewer, Thank you for the suggestions. We included your observations in the section “Conclusion and future work” (lines 536-550) as detailed as follow. 

"Validation and psychometric properties of self-reported questionnaires depend on the samples, when it has been applied, and its use [63,64]. In this sense, the evidence presented in this article is not definitive. Along the time, we expect to obtain different validity evidence, in which we will present more information about the strengths, weaknesses, and characteristics of the DFS-BR, in its long and short versions.

In further tests and validations of the DFS-BR, we will conduct studies with focus in the interpretation of the scale values for different groups or subjects. The interpretation of scale values depends also the context, and although we presented here validation for general activities, each response has a specific activity. These activities and others (that will be gathered in the future) will define the context in which we will analyze and interpret the gathered values for the dispositional flow scale. Establishing cut-off points to define what mean low or higher values of dispositional flow scale is another relevant future study that we expect to conduct in the future. This kind of study is also an important issue in the adaptation and construction of psychometric instrument that also depends on the population and context in which it is applied.

(… references …)

63. Price LR. Psychometric methods: Theory into practice. Guilford Publications;2016.

64. Sireci SG. On the validity of useless tests. Assessment in Education: Principles,Policy & Practice. 2016;23(2):226–235."

Comment 2 – Reviewer 3: There are some issues that, I believe, could contribute to improve the paper: ... 2. There were missing data? If yes, how they were treated (deleted, multiple imputation method,…).

Answer: Dear reviewer, missing and incomplete responses were automatically eliminated by the web application used by the participants to answer the items of the DFS-BR. We included this information in the section “Data analysis procedure” (lines 149-150) as detailed as follow. 

Incomplete responses were automatically removed from the dataset by the web application in which the respondents answered the items of the DFS-BR.

 

Comment 3 – Reviewer 3: There are some issues that, I believe, could contribute to improve the paper: ... 3. In the descriptive statistics the skewness and kurtosis should be reported in order to justify the estimation method used.

Answer: Dear reviewer, we added the indexes of skewness and kurtosis, as well as the statistics from Shapiro-Wilk test in the Table 1 as shown as follows.

Comment 4 – Reviewer 3: There are some issues that, I believe, could contribute to improve the paper: ... 4. If the authors would like report an index to justify the adequacy of the sample size, in CFA there is one specific global fit index for that purpose (Critical N). Given the statistical procedure follow, the lines 147 to 153 could be deleted.

Answer: Dear reviewer, we decided to not use the Critical N as part of fit indexes because it was not recommended by Hu, L. T., & Bentler, P. M. (1999). Cutoff criteria for fit indexes in covariance structure analysis: Conventional criteria versus new alternatives. Structural equation modeling: a multidisciplinary journal, 6(1), 1-55. Critical N indicates the small sample size at which chi-square would not be significant, but we reported (in the lines 150 to 155) the Kaiser-Meyer-Olkin and Bartlett’s test of spherity as measurement of adequacy to perform factor analysis. In other words, they are not fit indexes (as the Critical N), but the requirements to start the performing of CFA. 

Comment 5 – Reviewer 3: There are some issues that, I believe, could contribute to improve the paper: ... 5. As the authors do, the polychoric correlations should be used and, at least, a description of its adequacy should be commented. A good reference to do that is:

Holgado-Tello, F.P., Carrasco, M.A., Barrio, Mª.V., y Chacón, S. (2009). Factor Analysis of the Big Five Questionnaire using polychoric correlations in children. Quality & quantity, the International Journal of Methodology, 43(1), 75-85. If it is possible, the Bonferroni correction for the tests of adequacy of polychoric correlations should be reported.

Answer: Dear reviewer, we added a briefly explanation of why we decided to use the polychoric correlation in the section “Data analysis procedure” (lines 164-167) as described as follows.

"Polychoric correlation reduces the effect of attenuation that occur in the measured latent variable when it used a small number of item levels to estimate this value [32], and it is frequently applied for self-report psychometric instruments, such as personality trait [33].

…. (references)

32. Bonett DG, Price RM. Inferential methods for the tetrachoric correlation coefficient. Journal of Educational and Behavioral Statistics. 2005;30(2):213–225.

33. Holgado-Tello FP, Carrasco-Ortiz MA, del Barrio-Gándara MV, Chacón-Moscoso S. Factor analysis of the Big Five Questionnaire using polychoric correlations in children. Quality & Quantity. 2009;43(1):75–85. "

Answer: Dear reviewer, we also decided not to report the Bonferroni correlation as test of adequacy, because our decision to use polychoric correlation was based only in theoretical foundation (cited in the reference [33] and explained above).

Comment 6 – Reviewer 3: There are some issues that, I believe, could contribute to improve the paper: ... 6. The procedure followed to shorten the scale presents problems that should not be avoided. In this method, only reliability information is taken into account, and it has as a negative consequence that, if the reliability and discriminations is optimized, the validity suffers a deterioration. That is, the reliability increases at the cost of validity. This issue should be mentioned.

Answer: Dear reviewer, thanks for the important observation. In the new version of our paper, this validity threat, and the way in which we attempted to deal with it was added to the section “Construct validity of the DFS-Short BR” (lines 348-355) as indicated as follows.

"Notice that the method uses to define the alternative shorten version of the DFS-BR (with the model unidim2) increases the reliability of the instrument, but it may cause deterioration in the validity of the scale. To avoid a high degradation, improving simultaneously the reliability and the fit indexes, we intended to minimize the number of items that were changed from the model unidim1. We only changed the original item in the short version when the difference between the highest loading for each factor and the loading of the original item was greater than 0.20 in the multicorrelated model."

Comment 7 – Reviewer 3: There are some issues that, I believe, could contribute to improve the paper: ... 7. The adequacy of the model should be based on the global fit indexes. In this sense, if there are individual parameters that has not statistical relevance, but however, the global fit indexes are adequate, and they are conceptual congruence, there are not arguments to delete they from the model. Sometimes, the redaction in this sense is very restrictive. I could recommend taken into account, also, the sense of the items, and the global fit indices. In CFA perspective, the fit indexes should be related to the global of the model. In this sense, if the model fit, although there are individual parameters close to 0, or with low values, they could be considered relevant if they have coherence and congruence with the theoretical assumptions.

Answer: Dear reviewer, to avoid been restrictive, we rewrote some parts of the texts as detailed as follows. For the lines 256-259, we removed the details about the loadings of the item i35, and we justified its presence based on theoretical foundations. In the lines 385-388, we rewrote our previous observation making it to be like more suggestion that a need. The lines 403-406 has been rewrote to make it more like a suggestion that a requirement. In the lines 261-268, we removed the translations of each items to make the paragraph more readable that specific and tedious. 

Comment 8 – Reviewer 3: There are some issues that, I believe, could contribute to improve the paper: ...8. GFI should be reported.

Answer: Dear reviewer, thank you for your comment. We added the GIF (Goodness-of-Fit Index) and AGFI (Adjusted Goodness-of-Fit Index) to the Tables 2 and 6. A briefly explanation about these indexes and expected values was also added in the section “Data analysis procedure” (lines 178-181) as described as follows.

"(c) for the Goodness-of-Fit Index (GFI) and the Adjusted Goodness-of-Fit Index (AGFI), that takes into account the degrees of freedom of the model with respect to the number of variables, values close or greater than 0.90 were considered acceptable [34, 35];"

Comment 9 – Reviewer 3: There are some issues that, I believe, could contribute to improve the paper: ... 9. In general terms, the reliabilities of the scales are very low, this could be indicate that alternatives internal structures should be investigate.

Answer: Dear reviewer, thank you for your observation. We added your comment in the section “Conclusions and future works” (lines 490 to 492) as indicated as follows.

"However, the reliabilities in many of these factors are low (values of 0.06) which may be indication of the presence of internal structures that should be investigated in future studies."

---

## [Decision Letter · Decision Letter 2]

28 May 2021

Validation and psychometric properties of the Brazilian-Portuguese Dispositional Flow Scale 2 (DFS-BR)

PONE-D-20-30707R2

Dear Dr. Pinto,

We’re pleased to inform you that your manuscript has been judged scientifically suitable for publication and will be formally accepted for publication once it meets all outstanding technical requirements.

Kind regards,

Paolo Roma

Academic Editor

PLOS ONE

Additional Editor Comments (optional):

Reviewers' comments:

Reviewer's Responses to Questions

**Comments to the Author**

1. If the authors have adequately addressed your comments raised in a previous round of review and you feel that this manuscript is now acceptable for publication, you may indicate that here to bypass the “Comments to the Author” section, enter your conflict of interest statement in the “Confidential to Editor” section, and submit your "Accept" recommendation.

Reviewer #1: All comments have been addressed

Reviewer #3: All comments have been addressed

2. Is the manuscript technically sound, and do the data support the conclusions?

Reviewer #1: Yes

Reviewer #3: Yes

3. Has the statistical analysis been performed appropriately and rigorously? 

Reviewer #1: Yes

Reviewer #3: Yes

4. Have the authors made all data underlying the findings in their manuscript fully available?

Reviewer #1: Yes

Reviewer #3: Yes

5. Is the manuscript presented in an intelligible fashion and written in standard English?

Reviewer #1: Yes

Reviewer #3: Yes

6. Review Comments to the Author

Reviewer #1: (No Response)

Reviewer #3: Dear authors, congratulation for the result of the paper and good luck for forthcoming researches.

7. PLOS authors have the option to publish the peer review history of their article (what does this mean?). If published, this will include your full peer review and any attached files.

Reviewer #1: **Yes: **Renan Codonhato

Reviewer #3: No

---

## [Editor Report · Acceptance letter]

5 Jul 2021

PONE-D-20-30707R2 

Validation and psychometric properties of the Brazilian-Portuguese Dispositional Flow Scale 2 (DFS-BR) 

Dear Dr. Bittencourt:

I'm pleased to inform you that your manuscript has been deemed suitable for publication in PLOS ONE. Congratulations! Your manuscript is now with our production department. 

Kind regards, 

on behalf of

Prof. Paolo Roma 

Academic Editor

PLOS ONE